# Coexistence from a lion's perspective: Movements and habitat selection by African lions (*Panthera leo*) across a multi-use landscape

Ingela Jansson[1,2]* , Arielle W. Parsons[3] , Navinder J. Singh[1], Lisa Faust[3], Bernard M. Kissui[4], Ernest E. Mjingo[5], Camilla Sandström[6], Göran Spong[1]

1 Molecular Ecology Group, Department of Wildlife, Fish and Environmental Studies, Swedish University of Agricultural Sciences, Umeå, Sweden, 2 KopeLion, Ngorongoro Conservation Area, Arusha, Tanzania, 3 Alexander Center for Applied Population Biology, Lincoln Park Zoo, Chicago, Illinois, United States of America, 4 School for Field Studies, Karatu, Tanzania, 5 Tanzania Wildlife Research Institute, Arusha, Tanzania, 6 Department of Political Science, Umeå University, Umeå, Sweden

ʘ These authors contributed equally to this work.
* ingela.jansson@slu.se

**Data Availability Statement:** The one large datafile, containing all data for the analyses in this manuscript is available from the Dryad database

## Abstract

Diminishing wild space and population fragmentation are key drivers of large carnivore declines worldwide. The persistence of large carnivores in fragmented landscapes often depends on the ability of individuals to move between separated subpopulations for genetic exchange and recovery from stochastic events. Where separated by anthropogenic landscapes, subpopulations' connectivity hinges on the area's socio-ecological conditions for coexistence and dispersing individuals' behavioral choices. Using GPS-collars and resource- and step-selection functions, we explored African lion (*Panthera leo*) habitat selection and movement patterns to better understand lions' behavioral adjustments in a landscape shared with pastoralists. We conducted our study in the Ngorongoro Conservation Area (NCA), Tanzania, a multiuse rangeland, that connects the small, high density lion subpopulation of the Ngorongoro Crater with the extensive Serengeti lion population. Landscape use by pastoralists and their livestock in the NCA varies seasonally, driven by the availability of pasture, water, and disease avoidance. The most important factor for lion habitat selection was the amount of vegetation cover, but its importance varied with the distance to human settlements, season and time of day. Although we noted high levels of individual variation in tolerance for humans, in general lions avoided humans on the landscape and used more cover when closer to humans. Females showed more consistent avoidance of humans and stronger use of cover when near humans than did males. Connectivity of lion subpopulations does not appear to be blocked by sparse pastoralist settlements, and nomadic males, key to subpopulation connectivity, significantly avoided humans during the day, suggesting a behavioral strategy for conflict mitigation. These results are consistent with lions balancing risk from humans with exploitation of livestock by altering their behaviors to reduce potential conflict. Our study lends some optimism for the adaptive capacity of lions to promote coexistence with humans in shared landscapes.

(accession number https://doi.org/10.5061/dryad.j6q573nnb)

**Funding:** The project was supported by grant from the Swedish Research Council, awarded to GS [grant No. 2014-03382]. IJ also was awarded several grants over the years 2012-2023 for operations and equipment linked to this research: yearly grants from NABU International (https://en.nabu.de/about/index.html); National Geographic Society (https://www.nationalgeographic.org/society/) through the Waitt Foundation Grant awarded to C. Packer, and managed by IJ [grant No. W235-12]; National Geographic Society through the Big Cat Initiative [grant No. B15-15, and No. B4-17]; Panthera (years 2012, 2017-2019, https://panthera.org/); Wildaid (years 2017-2019, https://wildaid.org/); Lion Recovery Fund (https://lionrecoveryfund.org/) [grant No. TZ-KL-01, TZ-KL-02-LRF]; SOS – IUCN Save Our Species (https://iucnsos.org/) [grant No. 2021A-6]. The funders had no role in study design, data collection and analysis, decision to publish, or preparation of the manuscript.

**Competing interests:** The authors have declared that no competing interests exist.

## Introduction

Large carnivores range over vast areas, requiring a mix of habitats to fulfill their basic needs of food, shelter, and reproduction. In a world with diminishing wild spaces, carnivore ranges increasingly overlap with humans, impinging movements [1] and leading to conflicts and persecution [2]. In many cases, anthropogenic encroachment is extreme enough to fragment large carnivore populations into small, increasingly isolated subpopulations, jeopardizing their long-term viability and persistence [2–4]. Such populations are more likely to experience negative effects from stochastic events, genetic drift, and inbreeding, leading to lower levels of genetic variation. Ultimately, these effects increase extinction risk by enhancing vulnerability to environmental change and disease and/or reducing long-term health and fecundity [3].

The long-term persistence of large carnivores in fragmented landscapes often depends on the ability of individuals to disperse between isolated subpopulations [5]. However, navigating fragmented landscapes involves using and traversing habitats dominated by humans (i.e., shared landscapes). Shared landscapes may provide carnivores with ample resources (e.g., livestock) and reduced intra-guild competition [6] but also contain new risk-factors from direct conflict with people (e.g., persecution by hunting or poisoning), making them potential sink habitats [7]. Spatiotemporal scarcity of natural prey may also push carnivores to more risky behaviors, including increased overlap with people [8]. One of the main factors promoting successful connectivity for carnivores is management to enhance coexistence between humans and wildlife [9–11]. Large carnivore studies have frequently demonstrated increased mortality risk of dispersing individuals through fragmented landscapes, especially those occupied by humans [12]. However, efforts to make human-free corridors for wildlife movement are becoming less feasible as human populations grow globally and encroach on wild habitats [1]. Thus, finding ways to promote coexistence and facilitate connectivity through movement across human-occupied habitats is becoming increasingly important for the conservation of wildlife populations.

The African lion (*Panthera leo*) is adaptable to a variety of habitats, but soon ceases to persist in landscapes dominated by humans [13, 14]. Human land-uses compatible with naturally functioning landscapes, such as pastoralism, offer opportunities for coexistence with wildlife including lions [15, 16]. However, lions living among pastoralists are especially vulnerable to conflict through retaliatory killings, due to their habits of preying more on high value cattle (*Bos taurus*) than other livestock predators [17, 18], and lingering by the kill [18, 19] where other livestock predators, are more elusive [18]. In addition, traditional spearing of lions to demonstrate bravery is practiced among some pastoralist cultures, e.g. the Maasai [20, 21], contributing to human-lion conflict. The killings of lions by hunts and direct spearing, whether for retaliatory or ritual purpose, has undoubtedly shaped a relationship including both fear and respect between people and lions, a necessary trait for their coexistence [22, 23]. However, lions have an established capacity for flexibility, preferring natural prey over livestock where and when available [24] and exhibiting behavioral adaptations to avoid conflict with humans (e.g., becoming more nocturnal [9, 25]), often resulting in improved survival. Whether these behaviors can improve success in connecting isolated subpopulations remains unclear, however understanding the behavioral choices of lions in shared landscapes will be critical for making the best decisions for lion conservation as anthropogenic landscapes expand across Africa.

Lions are a group-living species with a complex social structure [26, 27]. Lions of different sexes and life stages behave differently, making unique behavioral choices based on landscape familiarity and response to threats [27, 28]. Female lions tend to be philopatric and territorial, typically settling adjacent to their natal area if they disperse [29–31]. In contrast, most males

depart from their natal pride at maturity, becoming nomadic until they are able to gain resident status with a pride. A male's resident state in a pride is temporary, typically 2–3 years, after which he is out-competed by other males or leaves voluntarily, re-entering a nomadic phase [27]. Nomadic males don't hold territories and are subordinate to resident males, wandering widely in search of females and often relegated to suboptimal habitats to avoid conflicts with other lions [27, 32]. Carnivore studies rarely consider these sex and life-stage differences when assessing habitat use and thus might miss important nuances of behavior and movements relevant to human-wildlife conflict.

Here, we present a study of the lion population of the Ngorongoro Conservation Area (NCA), Tanzania. The NCA is an important component of the Greater Serengeti Ecosystem and its only multiuse protected area, where traditional pastoralists and their livestock (cattle, sheep (*Ovis aries*), goats (*Capra hircus*), and donkeys (*Equus asinus*)) share the landscape with wildlife. The NCA's multiuse area connects the small, high density lion subpopulation of the Ngorongoro Crater with the extensive Serengeti National Park (NP). Increasing human activity and human-lion conflicts have been a main driver for lion population decline and disappearance across the NCA, halting dispersal and increasing the isolation of the Ngorongoro Crater subpopulation [21, 33]. Using data from GPS collars, we tested the best predictors of lion habitat use across the NCA at two spatial scales (local/landscape), within two seasons (wet/dry) and diel periods (night/day). To account for and understand differences between sexes and life-stages, we analyzed habitat use separately for resident lions of each sex, and for nomadic males. Our goal was to understand how lion habitat selection choices are affected by the benefits of ideal habitats and risks of human activity. We hypothesize that the intensity of anthropogenic presence reduces lion use of the shared landscape, especially for nomadic lions, but that lions mitigate risk through behavioral changes. Specifically, we predict that while environmental factors will be important to lion habitat selection, anthropogenic factors will have the largest associated effects. We also predict that the response of lions to humans will change at night and with natural habitat availability, with local-scale avoidance of people expected to be lower at night and in areas of ample cover for hiding, where and when the risk of detection by humans (and therefore conflict) is lower. Finally, we predict different responses to people during the wet season, depending on the scale. At the landscape-scale we expect higher avoidance of people due to more abundant and widespread natural prey (i.e., wet season migration), enabling lions to shift their range away from humans. However, at the local-scale (i.e. the immediate surrounding habitat) we expect lower avoidance of people during the wet season, because humans range closer to home and use the area around their settlement less intensely compared to the dry season, presenting a lower risk to lions.

## Methods

### Study area

The Ngorongoro Conservation Area (NCA) in Tanzania (-3˚E and 35˚N) is an 8,300 km$^2$ component of the Greater Serengeti Ecosystem. The NCA is unique in being a multiple-use protected area that allows traditional pastoralists and their livestock to coexist with wildlife, including lions [34]. People of the Maasai and Datooga (Barabaig) ethnic groups reside here at a population density of roughly 12 persons/km$^2$, of which the great majority subsist mostly or entirely on livestock. While most of the NCA is accessible for grazing and settlement, some areas are excluded for wildlife use only and photographic tourism: the craters (including the Ngorongoro Crater), the eastern highland forest, and a section called Ndutu near the Serengeti NP border (Fig 1). The landscape is a mix of forested highlands, rugged escarpments, volcanic calderas, *Vachellia*, *Senegalia* and *Commiphora* woodland savannah, and vast open grasslands.

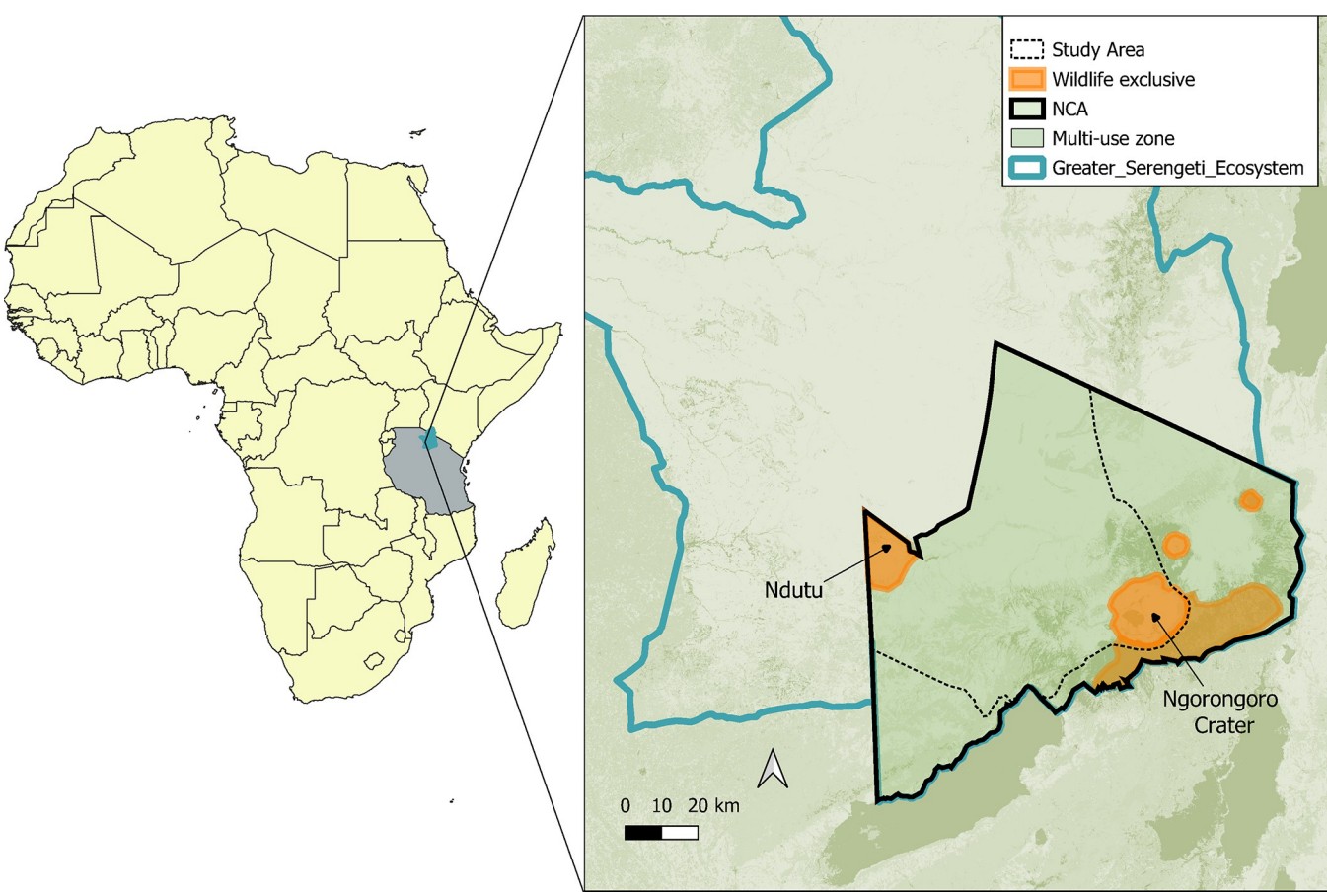

**Fig 1. Map of Ngorongoro Conservation Area and the Greater Serengeti Ecosystem, northern Tanzania.** Ngorongoro Conservation Area (NCA, black outline) is a large component of the Greater Serengeti Ecosystem (blue outline), Tanzania. While most of the NCA is multi-use, allowing pastoralists to coexist with wildlife, there are several areas reserved for wildlife only (in orange). The area of this study (dotted black line) encompassed around 4,800 km$^2$ of the central expanse of the NCA, from and including the Ngorongoro Crater westward to Ndutu and the border of Serengeti National Park to the north. Basemap reprinted from ESA WorldCover project 2020 under a CC BY 4.0 license.

Altitudes range between 1,000–3,000 meters above sea level, with a west to east rainfall gradient between 400–700 mm/year. The eastern highlands are foggy and cool during Jun-Aug, with nighttime temperatures often as low as +5°C. In the west, the vast short-grass plains form a seamless border to Serengeti NP–largely barren for much of the year, these plains provide highly nutritious pasture during the rains [35], a key driver of the great Serengeti ungulate migration [36]. NCA's varied topography, climate, vegetation, and seasons create a rich and heterogeneous landscape for abundant wildlife and provide valuable pastures for livestock. The area of this study encompassed approximately 4,800 km$^2$ of the central expanse of the NCA, from and including the Ngorongoro Crater westward to Ndutu and the border of Serengeti NP (Fig 1).

The NCA exhibits wet and dry seasonality, with a typical pattern of "short rains" Nov-Dec and "long rains" Mar-Apr [35]. The rains' time-lag effects on the pasture, a key resource for both wildlife and livestock, provides good wet-season grazing between Dec-May, and limited dry-season grazing between Jun-Nov. During the dry season, water is scarce, with most rivers being seasonal and many water-bodies undrinkable due to very high soda/salt levels. Consequently, wild herbivore abundance is low during the dry season and pastoralist movements intensify in search of pasture yet remain restricted to areas close to available water. Aside from

the herders moving between temporary homesteads, NCA's pastoralist families have become increasingly sedentary [37] and the majority reside in permanent homesteads. When the rains arrive, vast herds of migratory ungulates, including ~1.5 million wildebeest (*Connochaetes taurinus*) [38, 39], return to the shortgrass plains which covers ~40% of the NCA on its border to Serengeti. Between Jan–Apr, during the wildebeest's synchronized calving [40], pastoralists remove their cattle herds from the shortgrass plains to avoid infection from Malignant Catarrhal Fever, transmitted by the wildebeest's placenta to new-born calves [41, 42]. However, this virus does not affect sheep and goats, and some sparse pastoralist settlements remain. This natural phenomenon creates a period of greatly reduced overlap between pastoralists and their livestock and wildlife on the short-grass plains in the wet season, which then hosts a super-abundance of wildlife. Thus, lions residing on NCA's shortgrass plains experience a wet season with abundant natural conflict-free prey, and a dry season with high concentrations of risky prey (i.e., livestock) in the few areas with access to water. Consequently, we observe relatively more lion-livestock attacks during the dry season (S1 Fig), similar to other studies in the region finding increased attacks and/or risk when availability of native prey is low [19, 24, 43].

## Lion movements

To capture lion movements and use of habitat, we deployed satellite GPS collars (models GPS Plus or Vertex Lite by Vectronic Aerospace [44] on lions of both sexes. Between Oct. 2012-Mar. 2023, we had 22 different individuals collared: eight females (190 months total) and 14 males (252 months total). On average, individual lions were collared 19.9 months (range: 1.7 to 56.4 months) with nine lions being collared for two years or more (S1 Table). The collars had Iridium satellite data transmission (for regular data transfer) and VHF-beacon (for real-time manual tracking) and were scheduled to take positions every 1–2 hours continuously, day or night. The collar batteries lasted for 2–3.5 years, after which the collar was either removed (using remote drop-off) or replaced. We targeted lions for collaring from adult individuals of either sex from outside the Ngorongoro Crater, in the NCA multiuse area, from different groups and areas, based on our knowledge of the usual range of their pride or group. We prioritized collaring lions in areas with heightened risk of conflict. Apart from studying the lions' behavior through fine-scale movements, the purpose of the collars was to provide early-warning to livestock herders in the area, thus becoming an important tool for preventing human-lion conflicts, thereby enhancing the safety of the lions.

 **Ethical statement.** All research, fieldwork and data collection, including animal-handling to deploy collars on lions, complied with the Tanzania Wildlife Research Institute (Conduct of Wildlife Research) Regulations [45], and was carried out under the yearly renewed research permits granted to IJ with No's 2012-73-ER-90-15, 2013-147-NA-90-15, 2014-159-ER-2007-15, 2015-125-NA-2014-1165, 2016-229-NA-2014-165, 2017-243-NA-2007-15, 2018-362-NA-2014-165, 2019-341-NA-2006-79, 2020-256-NA-2019-065, 2021-548-NA-2019-065, 2022-771-NA-2019-068, and 2023-793-ER-2019-068 by the Tanzania Commission for Science and Technology (COSTECH; Dar es Salaam, Tanzania; rclearance@costech.or.tz) and Tanzania Wildlife Research Institute (TAWIRI; Arusha, Tanzania; researchclearance@tawiri.or.tz). A further permission to deploy collars was granted by NCA Authority, in letters No. NCAA/D/240/VOL.XXI/78, date 30/10/2012, to collar up to six lions simultaneously, and No. BD/158/711/01'E'/54, date 14/12/2021, permission renewed to collar up to eight lions simultaneously (NCAA; Ngorongoro Crater, Arusha, Tanzania; cc@ncaa.go.tz).

 In compliance with Tanzanian law, all lion captures and immobilizations for deployment or replacements of collars were performed by a NCA Authority or TAWIRI veterinarian. All collars featured a remote drop-off function, which could be activated via a timer or radio-

command using a release transmitter [44]. Therefore, re-capture of the lions was not necessary for collar removals. Collars were removed if they malfunctioned, batteries ran low, or the purpose for collaring that animal had been fulfilled.

Lions in NCA community lands are few and elusive, hence finding and capturing targeted individuals to deploy collars is challenging and done opportunistically, mainly at night, following observations of lions in the area. To attract lions closer to the vehicle for immobilization, we used a speaker (model Krakatoa, by FoxPro, USA) to broadcast a high-volume recording of feeding hyenas, a bleating buffalo calf, or the roars of a lion. To capture the lions, they were chemically immobilized with a drug mixture providing sedative, tranquilizing, and anesthetic effects (e.g., Zoletil and medetomidine), administered via a dart shot from a $CO_2$-powered dart-gun (Dan-Inject or Pneu-Dart) from a vehicle at a distance of 10–20 meters, following protocols described in [46]. Sedation time lasted typically less than one hour, during which time the veterinarian and the researcher monitored body conditions closely (breathing, temperature, circulation). The collars were fitted by an experienced field researcher supervised by the veterinarian. Afterwards we remained with the lion to closely monitor it until alert and deemed safe and well recovered after the immobilization.

Weighing approximately 1,200 g, the collars represented 1% or less of an adult lions' body weight and well within recommended limits [47]. We closely monitored all collared individuals through visual observations and field signs, and via daily checks of their movements as transmitted by the collars. We were observant of any signs of distress or negative short- or long-term impacts from the collars, including effects on reproduction, hunting and feeding behaviors, and social interactions. No apparent side effects from the collars were detected.

## Human activity proxy

To capture human activity across the landscape, our field team, recruited from the community residents based on their high local area knowledge, recorded GPS locations of bomas and water-points (i.e., rivers, springs, wells, and dams with drinkable water) within our study area between 2013–2016. Bomas are a pastoralist family homestead, with a large corral for livestock in the center surrounded by huts and occasionally small houses. Bomas are the center of livestock herding activity, where livestock depart every day for pasture and water, typically ranging up to 10 km per day from bomas. Boma locations included permanent and seasonal bomas (used only in the dry or wet season) with exact locations varying over time even as the general settlement areas have remained the same. Since the area used by nomadic male lions sometimes exceeded our main study area, we expanded the mapping of water-points and bomas to cover long-distance tracks by lions using a combination of remote mapping and ground surveys. For remote mapping, team members most familiar with the area viewed Google Earth satellite maps, assisted by artificially generated building locations [48], to geolocate bomas and assign them as permanent or seasonal. Using the same procedure, we mapped all buildings within the study-area. Out of the geolocated bomas, 33% were mapped in the field and of those 1,073 included a count of huts, with an average of 4.6 (sd = 3.44, median = 4, range 0–30) huts per boma. This included both large and small, permanent, and seasonal bomas, and provides a good representation of boma structures across the NCA. Compared to bomas, buildings are rare in the NCA and clustered within small areas of village centers, schools, hospitals, ranger posts, tourism lodges, and permanent camps. From a lion perspective, human activities from buildings are likely less relevant than bomas due to their lack of livestock activity, but nevertheless represent a concentrated threat from humans. The data on locations of bomas and buildings were combined, resulting in the spatial layer hereafter referred to as "bomas" that we used as a proxy for human activity.

## Landscape variables

We modelled lion habitat selection using landscape variables (Table 1) representing habitat characteristics and conflict risk (i.e., humans on the landscape). Habitat covariates included the percent of shrub/forest habitat within a 50m radius (hereafter "cover") to capture lion preference for sheltering and/or stalking under cover [49, 50]; a terrain ruggedness index (hereafter "TRI") [51] to capture lions' use of rugged, less accessible areas for shelter and/or stalking; an enhanced vegetation index (hereafter "EVI"; MODIS) to capture vegetation greenness, a proxy for forage quality and quantity [52], which tends to influence both wild and domestic herbivore (lion prey) distribution [53]; and the distance to the nearest riverbed (S2 Fig) which lions commonly use for stalking prey or resting in thick riverine vegetation [49, 54]. Although EVI values for the region are available monthly, to reduce computation time and model complexity, we summarized EVI for wet (Dec-May) and dry (Jun-Nov) seasons, respectively, by averaging over each year from 2012–2022. We used the month of April to represent the wet season and October to represent the dry season, typically the months with highest and lowest greenness, respectively, for each season [35]. Human-related covariates included the distance to the nearest boma, density of bomas within a defined radius (1km, 3km, 5km; see below) (S2 Fig), and the distance to the nearest water point. The bomas serve as proxy for human presence and intensity of human activity. The distance to nearest waterpoint is also related to human activity, as water is scarce in the NCA for much of the year, necessitating the use of limited naturally-fed water points for people, livestock and wildlife alike. However, preliminary tests indicated that distance to the nearest water point and distance to human activity were highly correlated (0.82), thus we selected only one variable, distance to human activity, for use in our models. Since many bomas shift seasonally, we modelled human related covariates separately in wet and dry seasons. All continuous variables were centered and scaled prior to analysis.

## Movement analysis

Habitat selection patterns are often scale-dependent [57], therefore we analyzed how lions select habitats at the landscape scale and how they then navigate those chosen habitats at the

**Table 1. Covariates used for modelling habitat selection of lions in the Ngorongoro Conservation Area, Tanzania.**

| Variable | Source | Resolution | Description |
|---|---|---|---|
| Cover | [48] | 50m | The percent of a 50m radius grid cell that contains $4m^2$ pixels classified as "Forest" or "Shrub", i.e. vegetation cover. |
| Dens_human | Field mapping, Google Earth, [48] | 1km, 3km and 5km. | Intensity of human activity (density of bomas) in 1km, 3km and 5km radius. Since pastoralists move bomas in the wet season, dry- and wet-season layers for each radius were produced. |
| Dist_human | Field mapping, Google Earth, [48] | 300m | Distance (m) to nearest human activity (boma). Since pastoralists move bomas out of certain areas in the wet season, dry- and wet-season layers were produced. |
| Dist_water | Field mapping | 300m | Distance (m) to nearest known water sources. Some water sources are only available in the wet season (we included only sources with water available > 2 months into the dry season), thus dry and wet season layers were produced. |
| Dist_river | [55], https://serengetidata.weebly.com/ and [56], https://www.hydrosheds.org/ | 300m | Distance (m) to the nearest river course. Rivers are dry most of the year but used by lions as a landscape feature for resting, ambush hunting, and denning. A single layer was produced and used in both dry and wet season modelling. |
| EVI | MODIS monthly vegetation Indices (MOD13A3) Version 6 | 1km | Enhanced Vegetation Index (EVI) represents vegetation cover while minimizing canopy background variation. Since EVI depends on precipitation, separate dry- and wet-layers were used. |
| TRI | Produced from USGS EROS digital elevation model using the terrain ruggedness index of [51] | 30-arc seconds | Terrain ruggedness index. Terrain ruggedness varies over the study area and more rugged terrain might provide more shelter for elusive lions. This index ranks the relative ruggedness of the landscape with higher numbers being more rugged. |

finer local scale. We assessed landscape-scale selection using resource selection functions (RSF), and fine-scale selection using step-selection functions (SSF). For each scale, we ran a separate set of models for three different lion categories: resident males (i.e., those settled with a pride and not making long-distance movements), nomadic males (i.e., males not associated with a pride, making frequent long-range movements) and females. Over the course of monitoring, males could be classified as nomadic or resident at different times, with 10 of the 15 males defined as nomadic during at least one month (S1 Table). To define a male lion's social status, we used the same criteria as in our long-term detailed lion monitoring with direct observations and individual recognition [26]. A male was considered nomadic when he departed from his natal pride (usually by 2–3 years of age), and/or when he ceased to be the resident male of a pride. A male was defined as a resident when consistently observed with a pride, actively engaging with the females and siring offspring. Resident males tend to spend most of their time with the pride (or among multiple prides if resident in several simultaneously), especially during the takeover and establishment phase, and when they have young cubs [26, 27]. This consistent behavior, coupled with our daily monitoring surveys and movement data from collared lions, enables us to characterize male status with confidence.

We began our movement analysis by testing the best radius over which to calculate human density (i.e., boma density). Bomas tend to be distributed widely across the landscape (mean distance between nearest neighbors = 3.23km), thus we selected an appropriate radius size to represent this spread (3km). However, we expected different lion categories to respond differently to bomas, based on differences in movement patterns and behavior [58]. Thus, we tested larger and smaller radii for each lion category (1km, 3km and 5km), running a single SSF for each radius for each lion category, that we ranked by Akaike Information Criterion (AIC) to select the most supported radius for subsequent modelling steps. We assessed average movement distances for each category of lion using mean squared displacement (MSD) [59, 60]. We modelled the difference in MSD between wet and dry seasons using a generalized linear model with log link to assess the season in which each category of lion moves most. We modelled habitat selection (RSF and SSF) separately for wet and dry seasons and nocturnal and diurnal movements [25]. We separated day- and night-time positions using the R package suncalc [61]. By modelling wet/dry season and daytime/nighttime movements separately, we allowed for different relationships between habitat selection and each covariate between seasons and times of day.

**Landscape scale habitat selection.** At the landscape scale, we restricted our RSF analysis to the study area over which we had a complete dataset for bomas, removing the few exploratory lion tracks that occurred outside of that area. We used package amt [62] in program R [63] to generate random available locations (n = 100 locations for each lion location) within the study area to compare to the locations actually used by each animal. To avoid pseudo-replication and the influence of lengthy resting locations, we resampled the position of each animal at 12-hour intervals. We included random intercepts and slopes for each individual animal to account for individual behavioral variation. We used the R package glmmTMB [64] to run our models as weighted logistic regressions, with weight set at 1000 and the variance of the random intercept term fixed at $10^3$ [65].

**Local scale habitat selection.** We used SSFs to quantify lions' habitat selection at the local scale by comparing habitat covariates at locations that the lions had visited with habitat covariates at a random set of available locations (i.e. the habitats available along animal movement tracks) [66]. The available locations for SSF modelling are conditioned on the last step taken by the animal, meaning each observed step is paired with some number of random steps, resulting in a stratified dataset [65]. Again we used the amt package [62] to estimate each

animal's movement parameters for step length and turn angle, then sampled from these parameter distributions to generate random available locations (n = 50) around each used location. We did not restrict this analysis to the study area, including long-distance tracks around which we had targeted additional collection of covariate data. We extracted covariate vectors for each location and used the Poisson formulation of conditional logistic regression to estimate associated slope coefficients [67]. We included random intercepts for each strata (i.e., set of used and available steps) and random slopes to allow for variation in individual responses to covariates [65]. We used the R package glmmTMB to run our models, fixing the variance of the random intercept for strata to $10^3$ [65].

For each lion category (female, resident male, nomadic male), we tested six sub-models. First, we tested a model including all predictors. Next, we tested the value of adding an interaction term to this fully-parameterized model representing the selection for cover relative to human activity (i.e., the distance to and intensity of human activity, respectively). Next, we tested a model including only habitat-related predictors against a model including only human activity predictors. Initial tests with fully-parameterized models indicated the percent of a 50m radius area that provided cover had a strong influence on step-selection for each lion category. We therefore tested two additional habitat models, one where cover was the sole predictor and another where all habitat predictors except for cover were included. Following model-selection, we ran the top model for each lion category separately for each season (wet and dry) and by time of day (day, night) to determine if relationships with habitat and human covariates differed.

We assessed the fit of our top model using used-habitat calibration plots (UHC) via the R package uhcplots [68]. First, we split data for each lion category into random training and testing halves and fit the respective top model using the training dataset. From the fitted model, we randomly sampled 1000 times from the distribution of each slope coefficient, selecting a new slope each time for each covariate in the model. We then used these simulated slope coefficients to generate predicted values. We generated density plots to compare observed and predicted covariate values. We then visually assessed how well observed and predicted covariate distributions overlapped as a representation of model fit [68].

**RSS and resistance mapping.** Since dispersal between lion sub-populations is typically done by males [29, 30] who tend to make initial long-distance movements after leaving their natal pride [28], and sometimes throughout their lives [26], we based our resistance mapping solely on nomadic males. We used relative selection strength (RSS) based on our SSF model to calculate habitat resistance for nomadic males across the study area [69]. RSS is the ratio of the probability of a used step being selected to the probability of an available step being selected and is given by the exponential of the estimated slopes in the step-selection model [69]. Habitat resistance is represented by 1-RSS, thus we first calculated RSS for our top model. Following the method of [69, 70], we generated a dataset of 200 equally spaced values within each covariate's scaled range. We generated separate ranges for covariates that varied between wet and dry seasons and calculated resistance separately for each season. We then estimated an RSS function (i.e., relative selection curve) for each covariate in each season, given by:

$$RSS_i = \Delta h_i(\beta_i)$$

where $\Delta h_i$ is the difference between the mock value of covariate $i$ at location $x_1$ and the average attribute value across the landscape (0 for scaled continuous covariates). $\beta_i$ is the slope coefficient estimate of covariate $i$ from our step-selection model for a given season. For simplicity, we did not model resistance separately by night/day.

The resulting season-specific RSS functions for each covariate were used to predict the resistance of each cell in the study area based on cell-specific covariate values for that season. First, we calculated the RSS value for each cell separately for each covariate, then we subtracted the RSS value from 1 (i.e., resistance) for each covariate in each cell of the study area. Finally, we summed the resistance values across all covariates for each cell to get an index of total cell resistance in that season. Thus, the resistance value $R$ at each cell $i$ in a season is given by:

$$R_i = \sum_{n=i}^{\infty} \exp(1 - (\alpha_0 + \alpha_x X_i))$$

where $\alpha_0$ and $\alpha_x$ are the intercept and slope of the RSS function for covariate $x$, respectively and $X_i$ is the value of covariate $x$ at cell $i$. For display purposes, we log-transformed resistance values when making our resistance maps.

## Results

We tracked a total of 22 collared lions a total of 437.7 months, including eight females and 15 males. Of the males, six were exclusively nomadic during the time they were monitored, five were exclusively resident with a pride and the remaining four spent some time in both categories (S1 Table). Some animals were more represented in each category than others, with >50% of data for females and resident males coming from two animals, respectively, and >50% of data for nomadic males coming from three animals (S1 Table). When we tested different radii (1km, 3km and 5km) for the calculation of intensity of human activity (i.e. boma density), we found support for the 1km radius for males (nomadic and resident) and 5 km for females (S2 Table). On average, males (MSD: nomadic males 132.01km, SE = 1.57, resident males 56.34km, SE = 2.42) moved more than females (MSD: 21.82km, SE = 0.08). Nomadic males and females moved more in the wet season and resident males moved more in the dry season (S3 Table).

### Landscape-scale habitat selection

At the landscape-scale, all lions selected for habitats with high cover, regardless of season (Fig 2). Females and resident males selected for areas closer to rivers and females selected for areas with higher greenness (EVI) during the dry season while resident males selected for flatter terrain during the wet season (TRI; Fig 2). Lions showed little significant landscape-scale selection for distances from human activity, with only females selecting for lower human densities and nomadic males selecting for higher distances away from humans, both during the dry season only (Fig 2).

**Local scale habitat selection.** For all three lion categories, a fully parameterized SSF model which included interactions between the cover and human activity was supported over all other models (S4 Table). UHC plots indicated a well-calibrated and representative model (S3–S5 Figs). Cover was the strongest predictor of step-selection for all lion categories, with all showing a positive relationship, but relationships varied with the distance to and intensity of human activity (Figs 3 and 4; S6 and S7 Figs). At mean levels of cover (30.1%), step selection was negatively associated with the density of humans for females and resident males, but not nomadic males, especially in the wet season for females and dry season for resident males (Fig 3; S7 Fig). However, when comparing between day and night, nomadic males had a significant positive relationship with the distance to humans (i.e. preferring to be farther from humans) at mean levels of cover, especially during the day, but females and resident males had no significant relationships with the distance to humans (Fig 3). Females more strongly selected for cover when closer to humans compared to when they were farther away, but this relationship

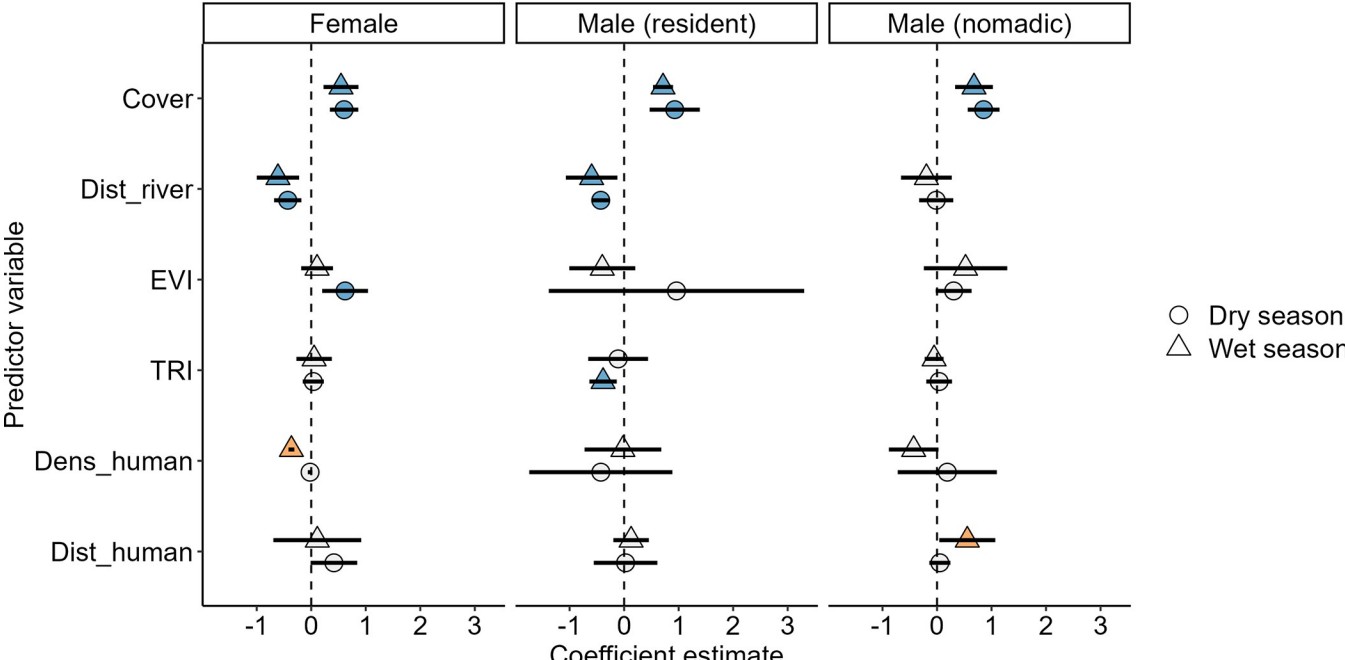

**Fig 2. Landscape-scale habitat selection by females, resident male and nomadic male lions.** Slope coefficients (points) and 95% confidence intervals (lines) associated with landscape-scale habitat selection (RSF) for three classes of lion (females, resident males and nomadic males) in the Ngorongoro Conservation Area, Tanzania. Blue points are habitat covariates and orange points are human-related covariates. Light grey points are those for which the 95% confidence interval overlaps zero. Dens_human represents intensity of human activity while Dist_human represents distance to human activity. Coefficients are estimated separately for seasons (wet and dry). All lions selected for habitats with high cover. Females and resident males selected for areas close to rivers. Females selected for areas of higher greenness (EVI) while resident males selected for flatter terrain (TRI), both during the dry season. Lions showed little landscape-scale selection for distances from human activity, with only females selecting for lower human densities and nomadic males selecting for higher distances away from humans, both during the wet season.

was only significant during the daytime and during the wet season (Figs 3–5). In contrast, resident males selected less strongly for cover when near humans during the wet season, with no such significant relationship during the dry season (Figs 3, 5). Nomadic males did not exhibit any significant relationships for the interactions between cover and distance to humans (Fig 3).

We noted significant negative relationships with the distance to rivers for all lion categories (i.e., preference for being close to rivers), significant in both season and dies periods except for nomadic males in the dry season and during the day (Fig 3). Resident males showed a strong negative relationship with TRI overall, being significantly negative during the dry season, but not during the wet season, and during the day but not at night (Fig 3). Nomadic males did not show any strong seasonal or daytime relationships with TRI but showed a significant negative relationship with TRI during the night (Fig 3). Females showed a significant positive relationship with TRI only during the dry season and during the day, but not at night or during the wet season (Fig 3). We did not note any significant relationship with greenness (EVI) except for resident males during the dry season and nomadic males at night, both positive relationships (Fig 3).

## RSS and resistance mapping

RSS was highest for cover for females and resident males (S8 Fig). For nomadic males, the two variables with the highest relative selection strength were EVI and the distance to human activity (S8 Fig). Relative resistance values across the study area were overall higher in the wet

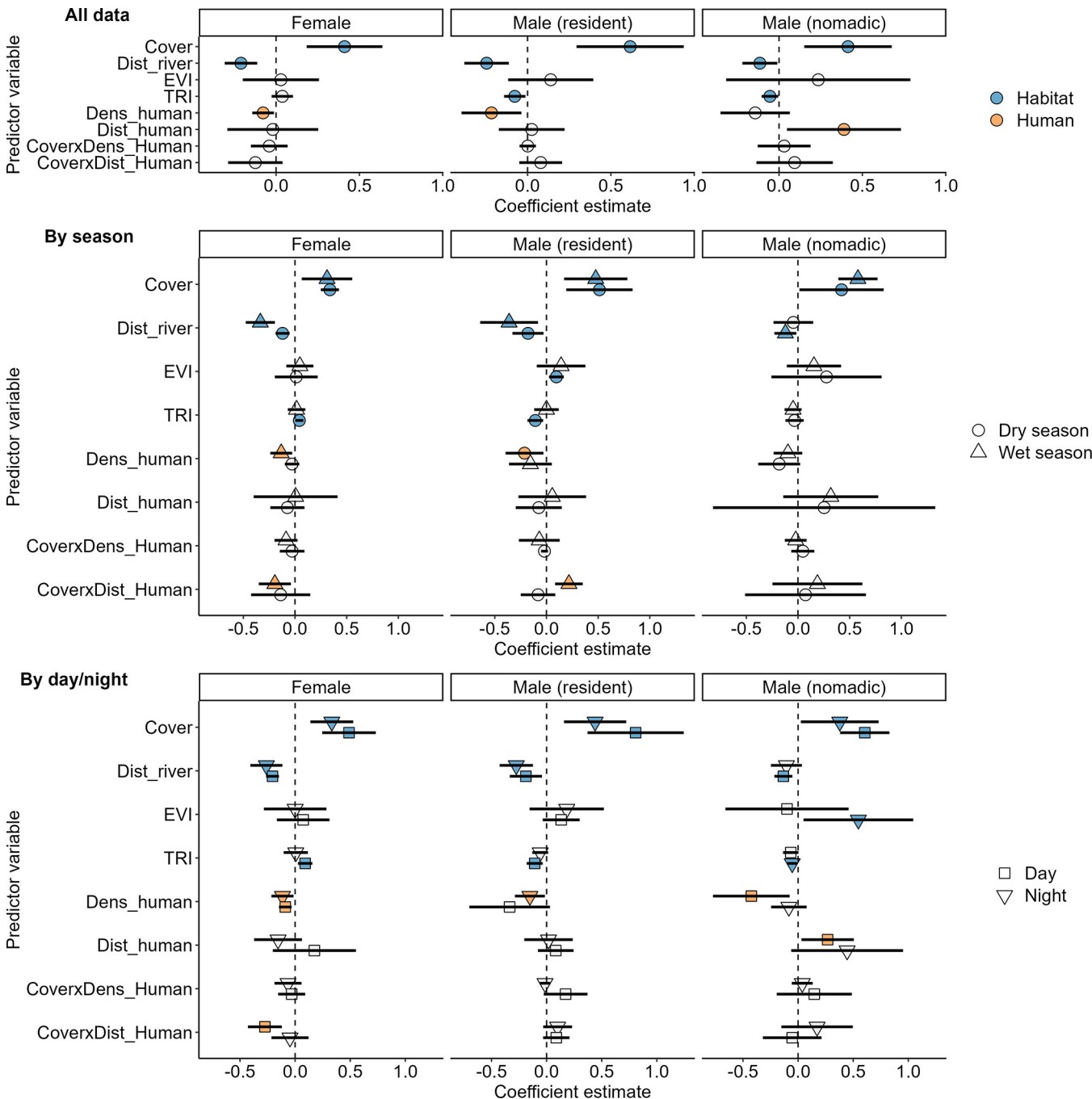

**Fig 3. Local-scale habitat selection by females, nomadic males and resident male lions.** Slope coefficients (points) and 95% confidence intervals (lines) associated with local-scale habitat selection (SSF) for three classes of lion (females, nomadic males and resident males) in the Ngorongoro Conservation Area, Tanzania. Blue points are habitat covariates, orange points are human-related covariates, light grey points are those for which 95% confidence interval overlap zero. Coefficients are estimated using all data points (top), then separately for seasons (wet and dry) and time of day (night and day). Dens_human represents intensity of human activity while Dist_human represents distance to human activity. The lions' habitat selection strengths varied with season and time of day, with generally less avoidance for humans in the dry season and at night.

season compared to the dry season (Fig 6). Around the Crater highlands and escarpment (southeastern block of the NCA), our resistance map presented a mosaic of areas of high and low resistance, where although village centers presented high resistance to lion movements,

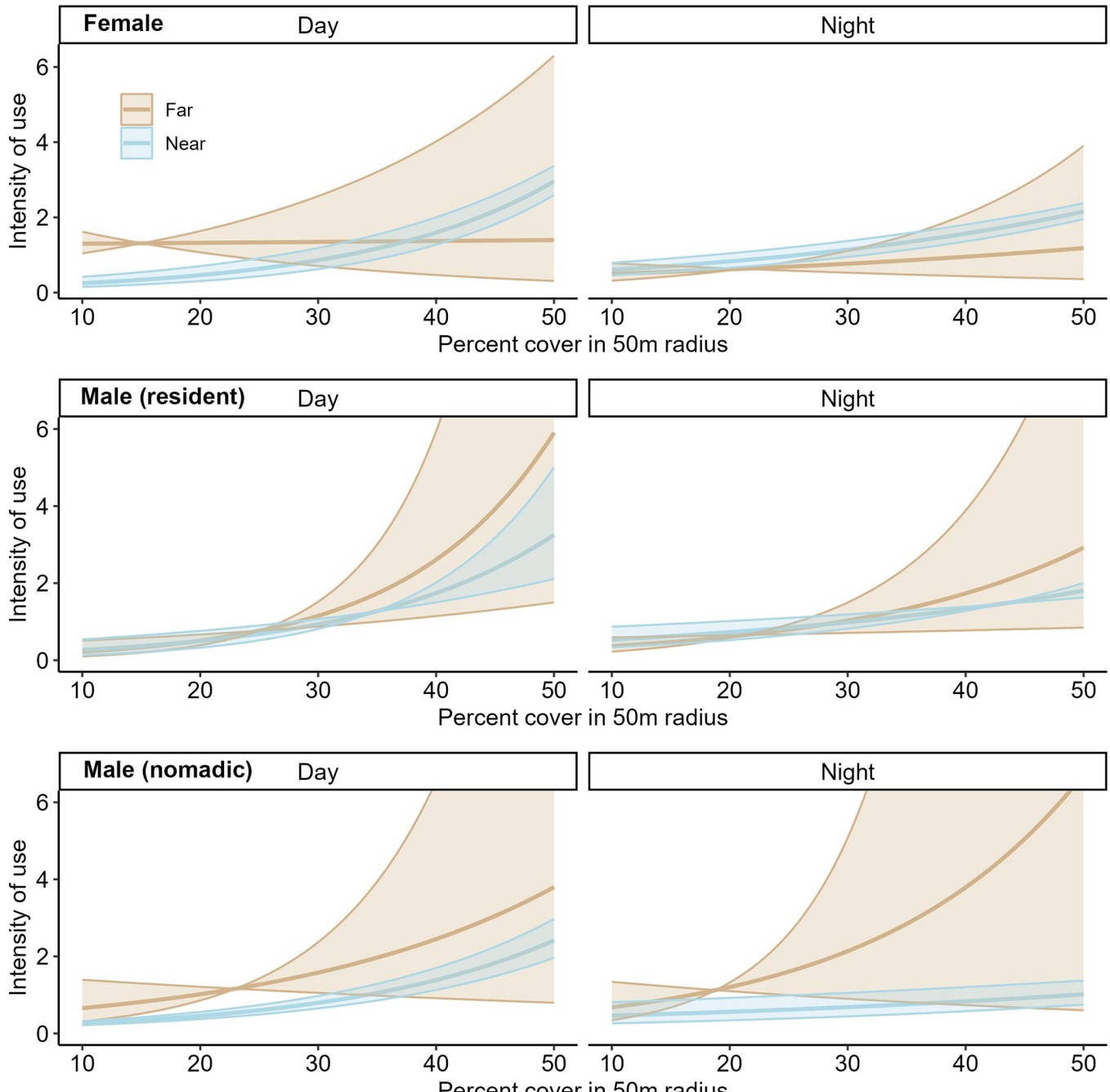

**Fig 4. Lion selection for cover when either far or near to humans between diel periods.** Local-scale selection (SSF) for cover when either far (2km) or near (500m) to humans (colors) based on the time of day (facets) for female, resident male and nomadic male lions. We have standardized the y-axis to the level of females for ease of interpretation, please note that means and 95% confidence intervals for resident and nomadic males extend beyond this level. All lions showed a positive relationship with the amount of cover, regardless of distance to humans. Females selected for significantly more cover near humans during the day, with no other statistically significant relationships observed.

these areas were small and interspersed with patches of low resistance habitat (Fig 6). The vast short-grass plains across the western part of the NCA has relatively little cover and sparse human settlements and represents an intermediate and homogenous level of resistance for

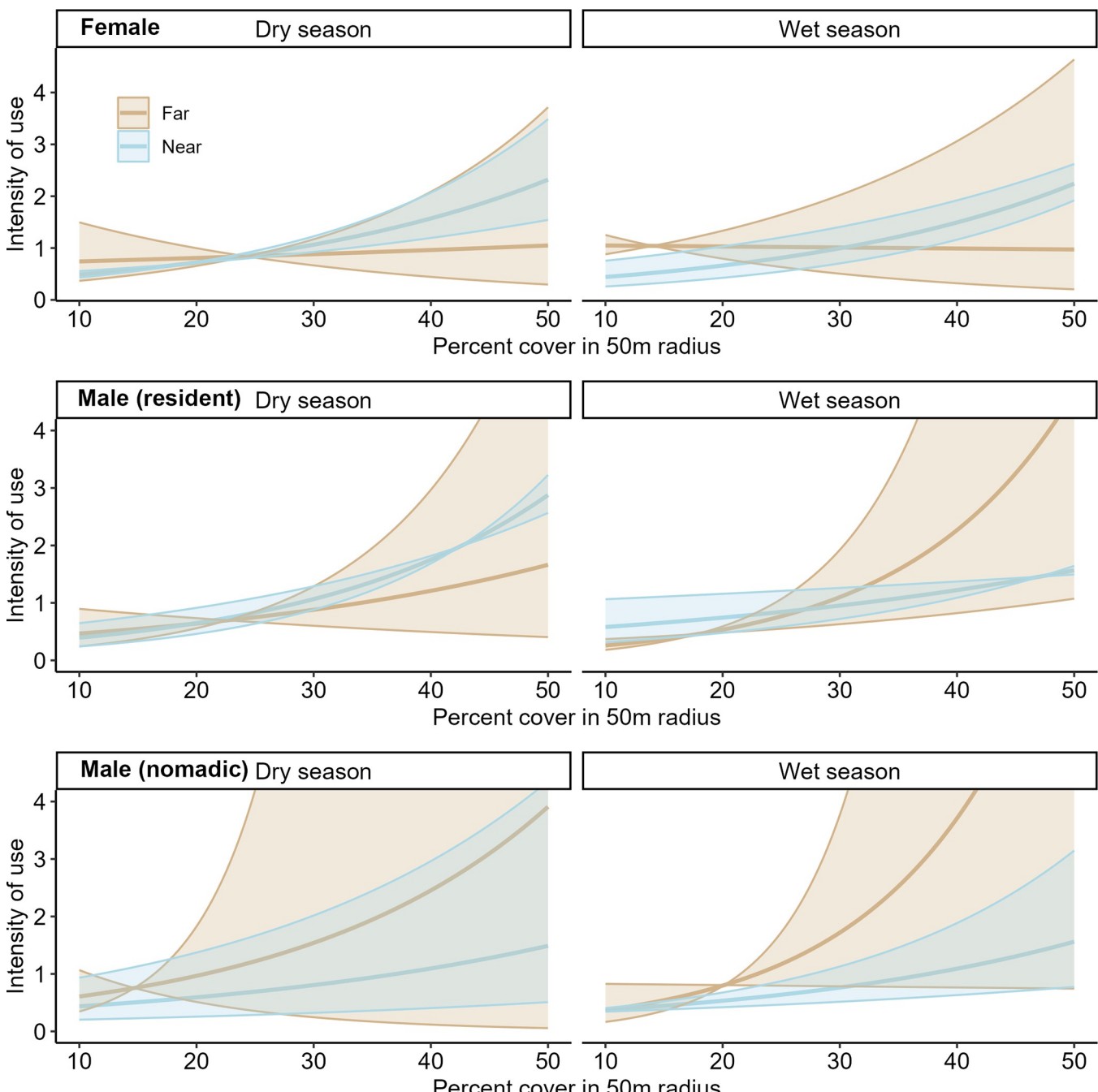

**Fig 5. Lion selection for degree of cover when either far or near to humans between seasons.** Local-scale selection (SSF) for cover when either far (2km) or near (500m) to humans (colors) based on season (facets) for female, resident male and nomadic male lions. We have standardized the y-axis to the level of females for ease of interpretation, please note that means and 95% confidence intervals for resident and nomadic males extend beyond this level. Females selected for significantly more cover near humans during the wet season and resident males selected for significantly less cover near humans during the wet season, with no other statistically significant relationships observed.

lions (Fig 6). A portion of the short-grass plains to the southwest of Ndutu is uninhabited by people, thus representing low resistance for lions (Fig 6). Despite the patchy distribution of low resistance habitats, our analysis did not detect any obvious major barriers to lion movements between the Crater and the neighboring Serengeti NP.

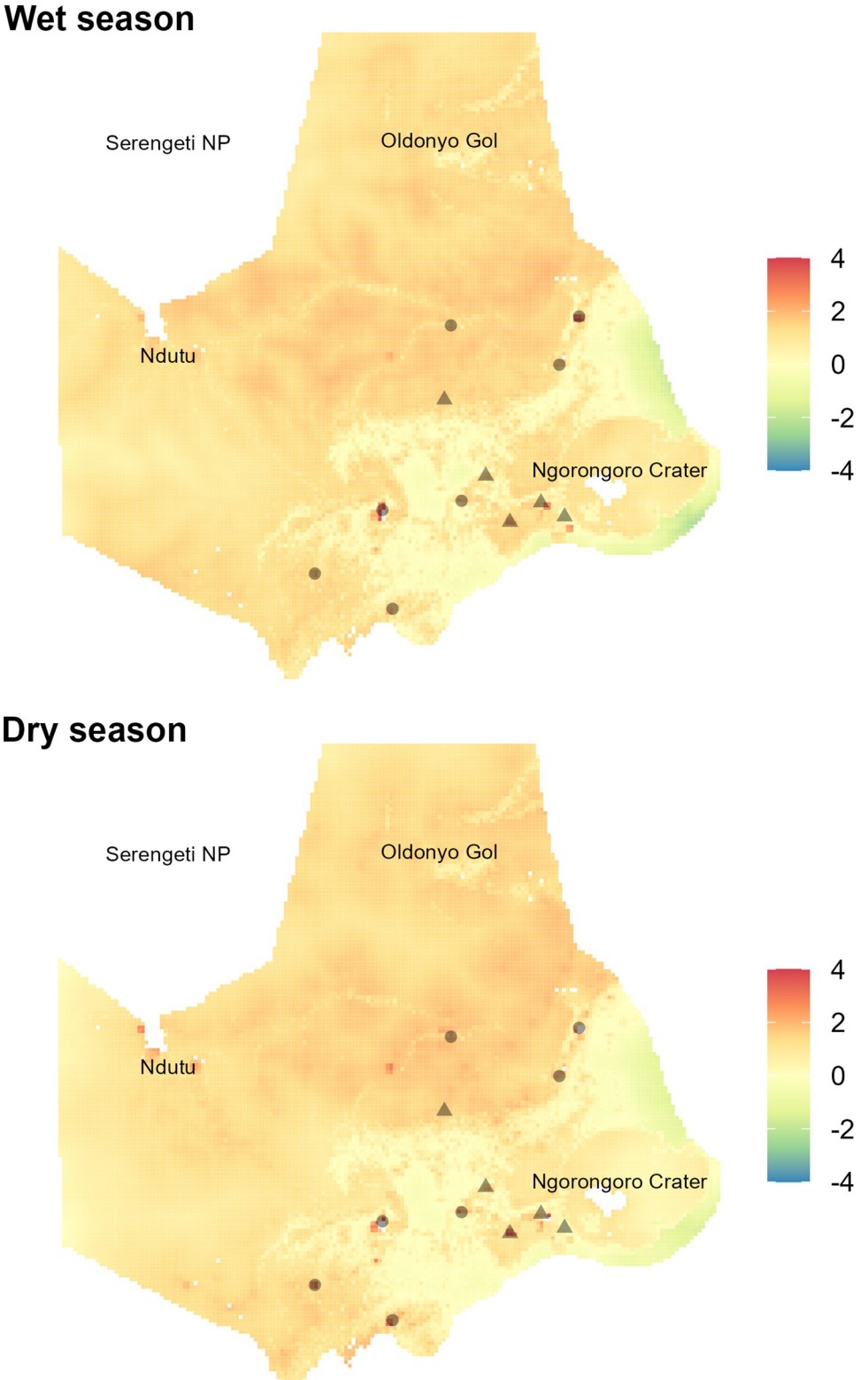

**Fig 6. Resistance map for nomadic male lions across NCA's multiuse landscape.** Resistance map for NCA nomadic males, which are not member of a pride and therefore likely in dispersal mode. Landscape resistance is assessed in both wet and dry seasons and show major regions, ward centers (circles) and villages (triangles). Relative resistance values, based on local-scale habitat selection model, SSF, are overall higher in the wet season (red = highest resistance/

avoidance, yellow = neutral, blue = least resistance/attraction). Relative resistance was generally highest in areas of dense human settlement, immediately to the west and north of the Ngorongoro Crater, and in steeper terrain. The areas of least resistance to lions occurred in forested habitats just north and south of these more intensely human-occupied areas.

## Discussion

We analyzed lion habitat selection at two scales to assess the relationship between habitat, the intensity of anthropogenic presence and lion use of a shared landscape. We found indications of strong selection for cover at both scales, with evidence of avoidance of humans, predominantly at the local scale. All lion categories significantly avoided areas of high human density or locations closer to humans. We found evidence of lions behaviorally avoiding humans through the use of cover when close to humans, and temporal adjustment to movements based on season and/or time of day. As wide-ranging carnivores increasingly inhabit fragmented landscapes under anthropogenic pressure, understanding how they respond to human presence and use suboptimal habitats will be critical for promoting dispersal between subpopulations and ensuring long-term large carnivore conservation.

Our prediction that human factors would have the largest effect on lion use of the shared landscape was partially supported. Although cover, not human activity, was the most important factor at both scales, our model selection supported the addition of an interaction term between these two factors. Coefficient estimates associated with the interaction between cover and human activity indicated lions, predominantly females and resident males, may use cover differently based on their proximity to humans. This is consistent with lions using cover to navigate the landscape, both to hunt prey [49, 50] and to avoid dangers, including conflict with humans [71]. At the local scale, cover was especially important for females, possibly because of their high philopatry, and thus familiarity with the pride's territory [27, 28]. This detailed knowledge of the local landscape may allow females to identify the best areas for hunting and more easily avoid risky situations compared to males. Additionally, the ability of females to move widely and avoid human-occupied areas is constrained by rearing young cubs, which might necessitate other behavioral strategies of avoidance, such as seeking cover.

Despite the importance of cover and human activity, relationships with human covariates were more uncertain than relationships with cover. This uncertainty is interesting, and is related to the use of random slopes in our model which account for variation in responses between individual lions (S9 and S10 Figs). At the landscape scale, the relatively high uncertainty related to humans indicates that some lions are more willing than others to live near humans, consistent with research showing individual attributes, such as boldness, facilitate the ability of some individual carnivores to settle in human-occupied areas [72, 73]. This was especially apparent for male lions, with relatively high variation in the response to the density of humans, especially during the dry season. Females, however, had markedly low variation in their significant negative response to human density in both seasons, suggesting that females are more consistent in their response to humans on the landscape compared to males. At the landscape scale, our prediction of higher avoidance of human activity in the wet season was supported for females and nomadic males, which also moved more during the wet season, while resident males responded similarly to humans in both seasons. The wet season's high abundance and wider distribution of migratory wild prey species in NCA reduces the need for lions to prey upon riskier livestock (i.e., risk-benefit tradeoff [8]), a pattern reflected in our monitoring of lion-livestock attack trends in NCA showing decreased lion-livestock attacks during the wet season (S1 Fig).

In the NCA, livestock are guarded constantly, herded to pasture during the day and returned to the relative safety of the bomas at night. To a lion, a boma represents both a danger and a potential food source to which they may be more willing to approach at night when the risk of conflict is reduced [71]. Avoidance of humans at the local scale varied with time of day and season, supporting our predictions. At mean levels of cover, nomadic males only significantly avoided humans during the day, consistent with other lion studies in the region [25, 71]. This suggests that males in wide-ranging exploratory modes (e.g., dispersal) are aware of humans on the landscape and making behavioral changes for conflict avoidance. This pattern was not noted for resident lions, with resident males only significantly avoiding humans at night and females showing significant avoidance of humans regardless of diel period. Furthermore, when females did come close to humans, they selected for more cover, especially during the daytime. This is consistent with the behavioral conflict avoidance observed in other carnivores (e.g., brown bear (*Ursus arctos*) [74] and spotted hyena [75]. However, despite their otherwise consistent avoidance of humans, females significantly avoided humans only during the wet season at both scales, not the dry season. This is consistent with our predictions at the landscape-scale, but at the local scale we predicted less avoidance of human activity during the wet season, when pastoralists use the landscape less intensively. Indeed, our prediction of less local-scale avoidance during the wet season was supported only for resident males, who significantly avoided humans during the dry season and selected less cover when near humans in the wet season. Nomadic males showed no significant seasonal avoidance patterns. Thus, while females appear to avoid humans whenever possible, even when the chance of encountering humans is low, males displayed less human avoidance with greater variation. This result may indicate less fear of humans by males, consistent with research showing that male lions, especially younger nomadic males, tend to be bolder and take more risks than females [5, 19, 32].

Our study demonstrates the importance of considering the behaviors of different sexes and behavioral states (i.e., resident/nomad) within different seasons and times of day for assessing wildlife responses to humans on the landscape. Our model of step-selection for nomadic males provides insight into the implications of humans on the landscape for movements of lions through the NCA, and connectivity between the NCA and neighboring lion populations (i.e., in Serengeti NP). This connectivity may be crucial for the long-term persistence of the lion sub-population in the Ngorongoro Crater, with its long history of close inbreeding [33, 76]. Our resistance map suggests no barriers to connectivity between lion subpopulations in the NCA at the current levels of human presence and activity. Indeed, our resistance map reflects our tracking data and lion observations [17] across the NCA which demonstrates the ability of lions to move through and occupy the multi-use landscape. While our study offers important insights, we note that our sample size is small, with 2–3 key individuals dominating the dataset for each lion category. While small sample sizes are typical of wildlife tracking studies, especially for large carnivores, our conclusions could have been influenced if those individuals were not representative of the rest of the population. We identify several other major areas where data collection could be improved to provide further value for understanding lion behavior and habitat selection in the NCA. First, we lacked key information on the abundance and distribution of natural prey, likely important factors driving lion movements. Although we noted positive selection for greenness (EVI) by resident males during the dry season and nomadic males at night, we cannot conclusively say whether this is related to prey abundance. Future studies that measure prey relative abundance across the NCA will be important to improving our understanding of the dynamics between lions, natural prey and humans as lions move through the landscape. Second, we used bomas as a proxy for human activity, which might be inadequate to capture human and livestock presence. Exact boma locations and especially livestock grazing areas, shift regularly. The latter is

particularly important in this context of human-lion conflict, as lions in the NCA tend to attack livestock at pasture [17]. Lastly, since our UHC plots demonstrated good fit for our models [68], we did not consider a non-linear relationship between lion step selection and distance to humans. However, since humans likely represent both a risk (retaliatory killings) and a benefit (source of livestock) to lions, nonlinearities in this relationship could exist that we have not explored.

We show that the human presence on the landscape can affect lion habitat use at multiple scales, with lions balancing risk through behavioral changes, consistent with other recent studies [8, 25, 71], including preferentially seeking cover habitats when approaching humans. These types of behavioral adaptations have been noted for other carnivores living near humans [74, 77] and exemplify the ways in which carnivore behavioral plasticity can promote coexistence with humans. However, humans remain dangerous to carnivores, with retaliatory killings still a major cause of lion mortality in NCA's multiuse landscape [17]. With the human population increasing in the NCA [78] and across the lion's range [13, 14], increased levels of conflict can be expected. Strategies to promote human-lion coexistence in the NCA are being implemented, including the Lion Guardians' model [79] where local pastoralist are employed to mitigate conflict, and monitoring and protect lions, as well as a trial of a conservation incentive payment program based on lion presence [80]. Early signs are encouraging that human tolerance for lions in the NCA can be increased [81], however the extent to which human behavioral plasticity can match lion behavioral choices to mitigate conflict remains to be seen. Environmental changes that negatively affect pasture productivity and/or wild prey abundance (e.g., climate change, drought, degradation of rangelands, invasive species) could be expected to increase levels of human-lion conflict. As human and lion ranges increasingly overlap across fragmented landscapes under changing environmental conditions, a better understanding of factors affecting lion habitat use and behavioral adjustments will be critical for the continued conservation of this iconic species in the NCA and throughout Africa.

## Supporting information

**S1 Table. Collared lions: Categories, collaring duration (months), and proportional representation by all lions and by its category.**
(DOCX)

**S2 Table. Test of the best radius over which to calculate the intensity of human activity on the landscape (i.e., boma density) for each lion category in Ngorongoro Conservation Area, Tanzania.**
(DOCX)

**S3 Table. Parameters estimated for the difference in mean squared displacement (MSD) of lion movements between wet and dry seasons.** The dry season is the reference level.
(DOCX)

**S4 Table. Model selection results for step-selection functions for female, resident male, and nomadic male lions in Ngorongoro Conservation Area, Tanzania between 2012–2023.** Dens_human represents intensity of human activity while Dist_human represents distance to human activity.
(DOCX)

**S1 Fig. Total number of attacks by lions on livestock in the study area (Fig 1) from 2015–2024, split between wet and dry-season attacks.** Livestock attacks were reported regularly by

local communities. When reports were received, locally trained field staff, with close ties to the pastoralist communities reporting the attacks, performed field visits to verify the attack and the type of predator involved.
(DOCX)

**S2 Fig.** Study area in the Ngorongoro Conservation Area showing the distribution of rivers (A) and bomas (B). The green polygon is the Ngorongoro Crater.
(DOCX)

**S3 Fig. UHC plots for the top step-selection model for nomadic male lions in Ngorongoro Conservation Area, Tanzania.** The observed distribution of each covariate at the presence points in the test dataset is given by the solid black lines, with associated 95% simulation envelope in gray. The available habitat across the landscape relative to each covariate is given in the dashed red line. Models were well-calibrated, with the observed distributions (solid black lines) falling predominantly within the gray simulation envelopes. Dens_human represents intensity of human activity while Dist_human represents distance to human activity.
(DOCX)

**S4 Fig. UHC plots for the top step-selection model for resident male lions in Ngorongoro Conservation Area, Tanzania.** The observed distribution of each covariate at the presence points in the test dataset is given by the solid black lines, with associated 95% simulation envelope in gray. The available habitat across the landscape relative to each covariate is given in the dashed red line. Models were well-calibrated, with the observed distributions (solid black lines) falling predominantly within the gray simulation envelopes. Dens_human represents intensity of human activity while Dist_human represents distance to human activity.
(DOCX)

**S5 Fig. UHC plots for the top step-selection model for female lions in Ngorongoro Conservation Area, Tanzania.** The observed distribution of each covariate at the presence points in the test dataset is given by the solid black lines, with associated 95% simulation envelope in gray. The available habitat across the landscape relative to each covariate is given in the dashed red line. Models were well-calibrated, with the observed distributions (solid black lines) falling predominantly within the gray simulation envelopes. Dens_human represents intensity of human activity while Dist_human represents distance to human activity.
(DOCX)

**S6 Fig. Selection for degree of forest/shrub cover when in areas of either a high (150/km2) or low (20/km2) density of humans (colors) based on the time of day (facets) for female, resident male and nomadic male lions.** We have standardized the y-axis to the level of females for ease of interpretation, please note that means and 95% confidence intervals for resident and nomadic males extend beyond this level.
(DOCX)

**S7 Fig. Selection for degree of forest/shrub cover when in areas of either a high (150/km2) or low (20/km2) density of humans (colors) based on the season (facets) for female, resident male and nomadic male lions.** We have standardized the y-axis to the level of females for ease of interpretation, please note that means and 95% confidence intervals for resident and nomadic males extend beyond this level.
(DOCX)

**S8 Fig. Relative selection strength for variables based on a local scale habitat selection model (SSF) for three classes of lion (females, nomadic males and resident males) in the**

**Ngorongoro Conservation Area, Tanzania.** Dens_human represents intensity of human activity while Dist_human represents distance to human activity. Relative selection strength was highest for habitat variable Cover (the percent of forest/shrub cover in a 50m radius) for females and resident males. For nomadic males, the variable with the highest relative selection strength was the distance to human activity.
(DOCX)

**S9 Fig. RSF without random slopes, showing much lower variation compared to when individual variation in relationships with predictors is accounted for (Fig 2).**
(DOCX)

**S10 Fig. SSF without random slopes, showing much lower variation compared to when individual variation in relationships with predictors is accounted for (Fig 3).**
(DOCX)

## Acknowledgments

This research was carried out under permissions granted by the Tanzania Commission for Science and Technology (COSTECH) and Tanzania Wildlife Research Institute (TAWIRI). We would like to thank the Government of Tanzania, TAWIRI, and Ngorongoro Conservation Area Authority for collaboration, and in particular their veterinarians, D. Wambura and the late A. Niaki. We would like to thank S. Capper, A. Estes, C. and L. Foley, and C. Packer for scientific advice on the study and manuscript. This work could not have been done without the KopeLion team, including R. Lelya for major input on lion monitoring and local ecological knowledge, and the NCA communities for guidance and participation.

## Author Contributions

**Conceptualization:** Ingela Jansson, Arielle W. Parsons, Navinder J. Singh, Lisa Faust, Bernard M. Kissui, Ernest E. Mjingo, Camilla Sandström, Göran Spong.

**Data curation:** Ingela Jansson, Arielle W. Parsons, Navinder J. Singh.

**Formal analysis:** Ingela Jansson, Arielle W. Parsons, Navinder J. Singh.

**Funding acquisition:** Ingela Jansson, Göran Spong.

**Investigation:** Ingela Jansson, Arielle W. Parsons, Navinder J. Singh.

**Methodology:** Ingela Jansson, Arielle W. Parsons, Navinder J. Singh.

**Project administration:** Ingela Jansson, Göran Spong.

**Supervision:** Lisa Faust, Camilla Sandström, Göran Spong.

**Writing – original draft:** Ingela Jansson, Arielle W. Parsons.

**Writing – review & editing:** Navinder J. Singh, Lisa Faust, Bernard M. Kissui, Ernest E. Mjingo, Camilla Sandström, Göran Spong.

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
