## [Decision Letter · Decision Letter 0]

4 Jul 2024

PONE-D-24-04716Coexistence from a lion’s perspective: movements and habitat selection by African lions (Panthera leo) across a multi-use landscapePLOS ONE

Dear Dr. Jansson,

Thank you for submitting your manuscript to PLOS ONE. After careful consideration, we feel that it has merit but does not fully meet PLOS ONE’s publication criteria as it currently stands. Therefore, we invite you to submit a revised version of the manuscript that addresses the points raised during the review process.

We look forward to receiving your revised manuscript.

Kind regards,

Bogdan Cristescu

Academic Editor

PLOS ONE

4. We note that Figures 1 and 6 in your submission contain [map/satellite] images which may be copyrighted. All PLOS content is published under the Creative Commons Attribution License (CC BY 4.0), which means that the manuscript, images, and Supporting Information files will be freely available online, and any third party is permitted to access, download, copy, distribute, and use these materials in any way, even commercially, with proper attribution. For these reasons, we cannot publish previously copyrighted maps or satellite images created using proprietary data, such as Google software (Google Maps, Street View, and Earth). For more information, see our copyright guidelines: http://journals.plos.org/plosone/s/licenses-and-copyright.

a. You may seek permission from the original copyright holder of Figures 1 and 6 to publish the content specifically under the CC BY 4.0 license. 

Additional Editor Comments:

First I want to apologize for the time it took to provide a decision on this manuscript, as it was challenging to find reviewers for it. The manuscript addresses an important topic and uses well-established techniques to accomplish its stated goals. The reviewers are generally supportive of the manuscript proceeding to publication, provided that some relatively major revisions are made, most substantively clarifying methodology and acknowledging sample sizes used. I also agree with Reviewer 1 that some of the figures are redundant in the main manuscript because they present relatively similar information.

In addition to their comments, please better clarify the classification of male lions in nomadic vs. resident - what thresholds were used to assign the individuals to one of these categories? E.g. how many days minimum did a male lion need to spend with a pride of females for it to be classified as resident? What type of interactions did the male need to have with the females and how did the different criteria factor in the classification? For example, how does a male that spent 30 days with a group of females but did not mate with them compare to a male that spent 3 days with the female but mated with them? Having clearly defined criteria would help not only this study but also classifications in future studies by other authors.

In the Methods you describe that local community members collected spatial data on water availability (river, springs, dams etc.). Please clarify if all these data were included in the modelling procedure. It appears that only distance to river was used for modelling, why not use all data that were collected? Especially given that you state rivers tend to be seasonal, whereas presumably the point water features are mostly permanent year-round.

There is a statement in the manuscript that in lion populations males are the ones that are important for connectivity, I would suggest nuancing this statement as females can also disperse although generally shorter distances. I am not suggesting running the resistance analysis on females also, but I do think that it would be important to mention in the text that females may also play a role in connectivity although perhaps less prominent than males.

Reviewers' comments:

Reviewer's Responses to Questions

**Comments to the Author**

1. Is the manuscript technically sound, and do the data support the conclusions?

Reviewer #1: Yes

Reviewer #2: Partly

2. Has the statistical analysis been performed appropriately and rigorously? 

Reviewer #1: Yes

Reviewer #2: I Don't Know

3. Have the authors made all data underlying the findings in their manuscript fully available?

Reviewer #1: Yes

Reviewer #2: Yes

4. Is the manuscript presented in an intelligible fashion and written in standard English?

Reviewer #1: Yes

Reviewer #2: Yes

5. Review Comments to the Author

Reviewer #1: Dear,

I have read the manuscript entitled “Coexistence from a lion’s perspective: movements and habitat selection by African lions (Panthera leo) across a multi-use landscape”. The study presents an analysis of lion habitat selection at two scales (landscape and local) in the Ngorongoro Conservation Area, as a function of human activity, in a conservation context aiming at reduce human-lion conflict. The manuscript is interesting and well written, the habitat selection methods are correctly applied, and I think it deserves to be published, despite the small sample size in each lion category (the authors can acknowledge this point in the discussion).

I think the method section deserves some clarifications, see my numerous minor comments along these lines below. Specifically, the authors should specify whether they scaled their continuous variables before running the SSF because of a risk of statistical problems for subsequent analyses, and justify why the resistance map varied between -1 and 1 (I don’t understand how is it possible with the current justification). I also strongly recommend to delete the Figure 5, which is redundant to Figure 3A and very misleading. To improve the general clarity of the manuscript, I suggest to improve figure presentation by shading non-significant results and facilitate a quick interpretation. In addition, the discussion is a little long and could be shortened and structured with sub-headings.

Minor comments:

L27. Replace « adaptations » by « adjustments ».

L131. Correct for “12 persons/km²”.

L280. I don’t think you run RSF and SSF with suncalc package. Please reformulate the sentence.

L284. Refer to the logistic regression in the section.

L293. Probably a mistake formulation here. There is no strata in a RSF. Muff et al. also recommend to use random slope coefficients to account for inter-individual variability in habitat selection, and weighted availability.

L298-300. The SSF definition is almost not understandable. I suggest something like “For SSF modelling, each observed step is paired with X random steps, resulting in a stratified dataset.”

L340. You have not mentioned that you scaled the continuous variables before adjusting the SSF. You cannot apply the regression coefficients to the new scaled variable if the coefficients were previously estimated using an unscaled variable.

L342-349. And what about the day/night differences?

L370. “In lieu of…” ? Maybe reformulate for “In absence of…”

L372-373. Add the distance unit.

L372-376. Add a Table presenting the regression summary in Appendix. Without information on the reference category for each categorical covariable, it is not possible to evaluate this part.

L388. What does the error bar represent?

L393-399. Delete.

L419. What does the error bar represent?

L425-426. Delete.

L432-433. Specify the chosen distance.

L433-438. Delete.

L442. Specify the value of the mean levels of cover (to the reader have a better idea of the local conditions).

L442-444. This results is also true for dry season. Unless you wish to state “both resident and nomadic males were significantly less likely to select locations closer to human activity during the wet season than dry season“. Authors should be clearer in their presentation of the results, specifying the basis for comparison. Verify also the rest of the manuscript.

L458-459. Results are already complex to follow due to crossing information on season, time of day and sex, and a little long. I suggest to delete this sentence (and maybe some others) to reduce a bit. To make it easier to read, I also suggest shading non-significant results in grey on the figures 2 and 3.

L462. I suggest to delete the Figure 5. Figure 5 is redundant with Figure 3A, although it shows the exponential of the estimates. In addition, all the explanations in this figure are redundant with the previous results section. You also use RSS to calculate map resistance only for nomadic males and at both seasons, but Figure 5 shows the results for the 3 lion categories without seasonal distinction, which is very misleading.

L481-484. Delete.

L486. Avoid to interpret result in the legend of the Figure 6 L490 and L491-494.

Figure 6. I don’t understand how the index of map resistance can range between -1 and 1, if it calculated from equation L358. The sum should vary between 0 and +inf

L499. Replace “regions” by “areas”.

L553-557. I’m not sure to correctly understand the justification for an artifact. Please reformulate.

L619-621. Do you have an idea of the number of lions killed annually?

L622-624. Can you develop an example in one sentence?

L633. Replace “adaptations” by “adjustments”.

The reviewer.

Reviewer #2: This paper makes great effort to improve our understanding of lion landscape and small-scale space use in the proximity of humans. Such an effort is commendable, and I support the publication of this work. However, I would like to make a couple of suggestions that might improve the manuscript.

The first issue that I would like to see captured in the text is a summarised version of Supplementary information S1. I think it is important that the reader is made aware that for resident males approximately 130 months of resident male data came from 8 animals and that almost half of the data came from only two animals. It is also important to note that 190 months of female data comes from 8 different animals but here more than half of the data are from only two animals. For the nomadic males 117 months of data were obtained from 9 animals but more than half of the data is from only 3 animals. By no means do I want to belittle the amount of effort and resources that went into obtaining the data used, but the question remains on how representative it is for all female, resident, and nomadic lions if for each group most of the data is from two or three individuals. I think this need to be explicitly shown and discussed. I do not think it change the conclusions, but the overall interpretation of results needs to be seen in this context. It will be interesting to see how well model results compare when the 2 or 3 key animals’ data are removed form the data set and the models re-run on the remaining data. A comparison between the full and partial model outcomes might be very revealing, but I do not insist on this for publication of this paper. Just please acknowledge the potential influence the individual preference of 2 or 3 key individuals could potentially have on the results.

The 2nd issue is the wild prey abundance issue. You recognise that an accurate idea of wild prey availability it is a crucial point of investigation for future studies but then you conclude that is important based on the assumption that EVI and season is correlated to wild prey abundance. This correlation might be true in the west on the short grass plains, but I am not convinced it holds for the entire area. My suggestion is to simply remove any text that try and correlate wet/dry season and EVI to prey abundance or at best state that it might be correlated. I made comments in text about this.

Another issue is that you deduct that lions do not need to predate on livestock during the wet season but that is in direct contradiction to the findings of a recent paper from the Tsavo area in Kenya. It might be worth having a look at the paper of Olivier, I. R., Tambling, C. J., Müller, L. & Radloff, F. G. T. (2023). Lion (Panthera leo) diet and cattle depredation on the Kuku Group Ranch Pastoralist area in southern Maasailand, Kenya. Wildlife Research, 50(4): 310-324.

A last point that is probably not necessary to address and might deviate a bit from the focus of the paper is the issue of retaliation and traditional killing of lion. In this and other East African landscapes, a landscape of fear exists for lion, you indirectly acknowledge this, created by the episodic killing of lion and the fashion within which this is done (I am not referring to poisoning but the active pursuit of lion by red clad men with spears). I often wonder how long it will take lion to lose there deep fear of Masai people when this practise is completely abandoned and following on this, how long it will take for lions to start killing people in a much higher frequency that is currently the case. Once that starts to happen this “co-existence” idea might not be all that feasible anymore as losing livestock is very different from losing human lives. I am thus of opinion that a more nuanced approach to retaliation and traditional killings is needed although I also at the same time understand the resentment for the cruelty involved.

Please find attached some minor comments in the accompanying file and good luck in getting this published.

6. PLOS authors have the option to publish the peer review history of their article (what does this mean?). If published, this will include your full peer review and any attached files.

Reviewer #1: No

Reviewer #2: No

---

## [Author Response · Author response to Decision Letter 0]

7 Aug 2024

Response to reviewers_PONE-D-24-04716

Editor comments:

Author response: Thanks for pointing this out. We have followed the instructions and made corrections accordingly. 

Author response: Data are now publicly accessible via this link: https://doi.org/doi:10.5061/dryad.j6q573nnb

Author response: Thanks for pointing this out. We have added an Ethical statement subsection under Methods. This section reads (line 186-222): 

“All research, fieldwork and data collection, including animal-handling to deploy collars on lions, complied with the Tanzania Wildlife Research Institute (Conduct of Wildlife Research) Regulations (Tanzania Wildlife Research Institute, 2020), and was carried out under the yearly renewed research permits granted to IJ with No’s 2012-73-ER-90-15, 2013-147-NA-90-15, 2014-159-ER-2007-15, 2015-125-NA-2014-1165, 2016-229-NA-2014-165, 2017-243-NA-2007-15, 2018-362-NA-2014-165, 2019-341-NA-2006-79, 2020-256-NA-2019-065, 2021-548-NA-2019-065, 2022-771-NA-2019-068, and 2023-793-ER-2019-068 by the Tanzania Commission for Science and Technology (COSTECH; Dar es Salaam, Tanzania; rclearance@costech.or.tz) and Tanzania Wildlife Research Institute (TAWIRI; Arusha, Tanzania; researchclearance@tawiri.or.tz). A further permission to deploy collars was granted by NCA Authority, in letters No. NCAA/D/240/VOL.XXI/78, date 30/10/2012, to collar up to six lions simultaneously, and No. BD/158/711/01'E'/54, date 14/12/2021, permission renewed to collar up to eight lions simultaneously (NCAA; Ngorongoro Crater, Arusha, Tanzania; cc@ncaa.go.tz). 

In compliance with Tanzanian law, all lion captures and immobilizations for deployment or replacements of collars were performed by a NCA Authority or TAWIRI veterinarian. All collars featured a remote drop-off function, which could be activated via a timer or radio-command using a release transmitter (VECTRONIC Aerospace, 2024). Therefore, re-capture of the lions was not necessary for collar removals. Collars were removed if they malfunctioned, batteries ran low, or the purpose for collaring that animal had been fulfilled. 

Lions in NCA community lands are few and elusive, hence finding and capturing targeted individuals to deploy collars is challenging and done opportunistically, mainly at night, following observations of lions in the area. To attract lions closer to the vehicle for immobilization, we used a speaker (model Krakatoa, by FoxPro, USA) to broadcast a high-volume recording of feeding hyenas, a bleating buffalo calf, or the roars of a lion. To capture the lions, they were chemically immobilized with a drug mixture providing sedative, tranquilizing, and anesthetic effects (e.g., Zoletil and medetomidine), administered via a dart shot from a CO2-powered dart-gun (Dan-Inject or Pneu-Dart) from a vehicle at a distance of 10-20 meters, following protocols described in (Kock and Burroughs, 2012). Sedation time was typically less than one hour, during which time the veterinarian and the researcher monitored body conditions closely (breathing, temperature, circulation). The collars were fitted by an experienced field researcher supervised by the veterinarian. Afterwards we remained with the lion to closely monitor it until alert and deemed safe and well recovered after the immobilization. 

Weighing approximately 1,200 g, the collars represented 1% or less of an adult lions’ body weight and well within recommended limits (REF Wilson 2021). We closely monitored all collared individuals through visual observations and field signs, and via daily checks of their movements as transmitted by the collars. We were observant of any signs of distress or negative short- or long-term impacts from the collars, including effects on reproduction, hunting and feeding behaviors, and social interactions. No apparent side effects from the collars were detected.”

4. We note that Figures 1 and 6 in your submission contain [map/satellite] images which may be copyrighted. All PLOS content is published under the Creative Commons Attribution License (CC BY 4.0), which means that the manuscript, images, and Supporting Information files will be freely available online, and any third party is permitted to access, download, copy, distribute, and use these materials in any way, even commercially, with proper attribution. For these reasons, we cannot publish previously copyrighted maps or satellite images created using proprietary data, such as Google software (Google Maps, Street View, and Earth). For more information, see our copyright guidelines: http://journals.plos.org/plosone/s/licenses-and-copyright.

Author response: For Figure 1, we have replaced the basemap to one that is open-access/use and added the following to the figure legend (line 146): “Basemap reprinted from ESA WorldCover project 2020 under a CC BY 4.0 license”. Figure 6 does not contain any satellite imagery; its contents were created solely by the authors. 

5. First I want to apologize for the time it took to provide a decision on this manuscript, as it was challenging to find reviewers for it. The manuscript addresses an important topic and uses well-established techniques to accomplish its stated goals. The reviewers are generally supportive of the manuscript proceeding to publication, provided that some relatively major revisions are made, most substantively clarifying methodology and acknowledging sample sizes used. I also agree with Reviewer 1 that some of the figures are redundant in the main manuscript because they present relatively similar information.

Author response: Thanks, we are both thrilled and relieved to hear back from you. We have addressed all of the reviewer comments below. These comments are a great help for us to improve on the manuscript.

6. In addition to their comments, please better clarify the classification of male lions in nomadic vs. resident - what thresholds were used to assign the individuals to one of these categories? E.g. how many days minimum did a male lion need to spend with a pride of females for it to be classified as resident? What type of interactions did the male need to have with the females and how did the different criteria factor in the classification? For example, how does a male that spent 30 days with a group of females but did not mate with them compare to a male that spent 3 days with the female but mated with them? Having clearly defined criteria would help not only this study but also classifications in future studies by other authors.

Author response: A good point to have this better clarified. The amended text now include how we used observations of social interactions for our definitions, and the text now reads (lines 284-292): “To define a male lion's social status, we used the same criteria as in our long-term detailed lion monitoring with direct observations and individual recognition (Packer, 2023). A male was considered nomadic when he departed from his natal pride (usually by 2-3 years of age), and/or when he ceased to be the resident male of a pride. A male was defined as a resident when consistently observed with a pride, actively engaging with the females and siring offspring. Resident males tend to spend most of their time with the pride (or among multiple prides if resident in several simultaneously), especially during the takeover and establishment phase, and when they have young cubs (Packer, 2023; Pusey and Packer, 1987). This consistent behavior, coupled with our daily monitoring surveys and movement data from collared lions, enabled us to characterize male status with confidence.”

7. In the Methods you describe that local community members collected spatial data on water availability (river, springs, dams etc.). Please clarify if all these data were included in the modelling procedure. It appears that only distance to river was used for modelling, why not use all data that were collected? Especially given that you state rivers tend to be seasonal, whereas presumably the point water features are mostly permanent year-round.

Author response: Yes, we did include these “water points” in the preliminary analyses – but as they were highly correlated with the boma locations, we removed them. This appears in the text in the first paragraph of the Methods (lines 266-268): “However, preliminary tests indicated that distance to the nearest water point and distance to human activity were highly correlated (0.82), thus we selected only one variable, distance to human activity, for use in our models.” To avoid confusion, we have removed “water points” from Table 1.

8. There is a statement in the manuscript that in lion populations males are the ones that are important for connectivity, I would suggest nuancing this statement as females can also disperse although generally shorter distances. I am not suggesting running the resistance analysis on females also, but I do think that it would be important to mention in the text that females may also play a role in connectivity although perhaps less prominent than males.

Author response: Thanks, that is indeed true. We have amended this to read (Lines 85-86): "Females lions tend to be philopatric and territorial, typically settling adjacent to their natal area if they disperse (Curry et al., 2019; Dolrenry et al., 2014; VanderWaal et al., 2009). In contrast, most males depart from their natal prides at maturity..."

Further, (Lines 351-353) "Since dispersal between lion sub-populations is typically done by males (Curry et al., 2019; Dolrenry et al., 2014) who tend to make initial long-distance movements after leaving their natal pride (Elliot et al., 2014), and sometimes throughout their lives (Packer, 2023), we based our resistance mapping solely on nomadic males." 

Reviewer #1

9. I have read the manuscript entitled “Coexistence from a lion’s perspective: movements and habitat selection by African lions (Panthera leo) across a multi-use landscape”. The study presents an analysis of lion habitat selection at two scales (landscape and local) in the Ngorongoro Conservation Area, as a function of human activity, in a conservation context aiming at reduce human-lion conflict. The manuscript is interesting and well written, the habitat selection methods are correctly applied, and I think it deserves to be published, despite the small sample size in each lion category (the authors can acknowledge this point in the discussion).

Author response: Thank you! We have added an acknowledgement of sample size to the discussion (Lines 563-567), “While our study offers important insights, we note that our sample size is small, with 2-3 key individuals dominating the dataset for each lion category. While small sample sizes are typical of wildlife tracking studies, especially for large carnivores, our conclusions could have been influenced if those individuals were not representative of the rest of the population.” In addition, we have added the proportion of location data that each lion contributed, overall and within its category in the S1 Table. 

I think the method section deserves some clarifications, see my numerous minor comments along these lines below. Specifically, the authors should specify whether they scaled their continuous variables before running the SSF because of a risk of statistical problems for subsequent analyses, and justify why the resistance map varied between -1 and 1 (I don’t understand how is it possible with the current justification). I also strongly recommend to delete the Figure 5, which is redundant to Figure 3A and very misleading. To improve the general clarity of the manuscript, I suggest to improve figure presentation by shading non-significant results and facilitate a quick interpretation. In addition, the discussion is a little long and could be shortened and structured with sub-headings.

Author response: Thank you for good and valid evaluations and comments. We have removed Figure 5, shortened the Discussion and added the suggested shading to Figures 2 and 3.

10. L27. Replace « adaptations » by « adjustments ».

Author response: This has been done.

11. L131. Correct for “12 persons/km²”.

Author response: This has been done.

12. L280. I don’t think you run RSF and SSF with suncalc package. Please reformulate the sentence.

Author response: We have rewritten this to read (lines 303-304): “We separated day- and night-time positions using the R package suncalc”

13. L284. Refer to the logistic regression in the section.

Author response: This has been added.

14. L293. Probably a mistake formulation here. There is no strata in a RSF. Muff et al. also recommend to use random slope coefficients to account for inter-individual variability in habitat selection, and weighted availability.

Author response: Thank you, the references to strata was indeed a typo here and has been removed. We also appreciate the recommendation to use random slopes in our models and a weighted likelihood for the RSF, we have implemented these changes and updated the methods to reflect that. As would be expected, this has decreased coefficient confidence for many relationships, and we have updated the results and discussion to reflect those changes.

15. L298-300. The SSF definition is almost not understandable. I suggest something like “For SSF modelling, each observed step is paired with X random steps, resulting in a stratified dataset.”

Author response: Thank you, we have made the suggested change. This sentence now reads (Lines 321-323): “The available locations for SSF modelling are conditioned on the last step taken by the animal, meaning each observed step is paired with some number of random steps, resulting in a stratified dataset [52].”

16. L340. You have not mentioned that you scaled the continuous variables before adjusting the SSF. You cannot apply the regression coefficients to the new scaled variable if the coefficients were previously estimated using an unscaled variable.

Author response: Thank you for catching that! We did indeed center and scale prior to analysis and have added a sentence to that effect (Line 270): “All continuous variables were centered and scaled prior to analysis.” 

17. L342-349. And what about the day/night differences?

Author response: We did not assess day/night differences in resistance, instead focusing on seasonal differences (thus averaging over day/night). We have clarified that in the text (Line 367-368): “For simplicity, we did not model resistance separately by night/day.” 

18. L370. “In lieu of…” ? Maybe reformulate for “In absence of…”

Author response: We have made the su

---

## [Decision Letter · Decision Letter 1]

16 Sep 2024

Coexistence from a lion’s perspective: movements and habitat selection by African lions (*Panthera leo*) across a multi-use landscape

PONE-D-24-04716R1

Dear Dr. Jansson,

We’re pleased to inform you that your manuscript has been judged scientifically suitable for publication and will be formally accepted for publication once it meets all outstanding technical requirements.

Kind regards,

Paulo Corti, Ph.D.

Academic Editor

PLOS ONE

Additional Editor Comments (optional):

Reviewers' comments:

Reviewer's Responses to Questions

**Comments to the Author**

1. If the authors have adequately addressed your comments raised in a previous round of review and you feel that this manuscript is now acceptable for publication, you may indicate that here to bypass the “Comments to the Author” section, enter your conflict of interest statement in the “Confidential to Editor” section, and submit your "Accept" recommendation.

Reviewer #1: All comments have been addressed

2. Is the manuscript technically sound, and do the data support the conclusions?

Reviewer #1: (No Response)

3. Has the statistical analysis been performed appropriately and rigorously? 

Reviewer #1: (No Response)

4. Have the authors made all data underlying the findings in their manuscript fully available?

Reviewer #1: (No Response)

5. Is the manuscript presented in an intelligible fashion and written in standard English?

Reviewer #1: (No Response)

6. Review Comments to the Author

Reviewer #1: Dear,

Thank you for your thorough review of the manuscript and to have reanalyzed the statistical models. Authors have correctly include my recommendations. The ms is now clear and ready for publication.

I have just few minor comments. The line numbers are based on the manuscript in track change mode:

- L198: A parenthesis is missing

- L365: Correct for "with weight set at 1000 for random locations and..."

- L 512: Did you say "diel periods" rather than "dies periods"?

- L618-619 and 623-625 are partially redundant.

Sincerely,

The Reviewer

7. PLOS authors have the option to publish the peer review history of their article (what does this mean?). If published, this will include your full peer review and any attached files.

Reviewer #1: No

---

## [Editor Report · Acceptance letter]

24 Sep 2024

PONE-D-24-04716R1 

PLOS ONE

Dear Dr. Jansson, 

I'm pleased to inform you that your manuscript has been deemed suitable for publication in PLOS ONE. Congratulations! Your manuscript is now being handed over to our production team.

Kind regards, 

on behalf of

Dr. Paulo Corti 

Academic Editor

PLOS ONE